# Laying Hens: Why Smothering and Not Surviving?—A Literature Review

**DOI:** 10.3390/ani14111518

**Published:** 2024-05-21

**Authors:** Caroline Citta Mazocco, Sérgio Luís de Castro Júnior, Robson Mateus Freitas Silveira, Rosangela Poletto, Iran José Oliveira da Silva

**Affiliations:** 1Núcleo de Pesquisa em Ambiência (NUPEA), Escola Superior de Agricultura ‘‘Luiz de Queiroz’’ (ESALQ), Universidade de São Paulo (USP), Piracicaba 13418-900, SP, Brazil; sergio.castro@usp.br (S.L.d.C.J.); robsonsilveira@usp.br (R.M.F.S.); iranoliveira@usp.br (I.J.O.d.S.); 2Instituto Federal de Educação, Ciência e Tecnologia do Rio Grande do Sul (IFRS)-Campus Sertão, Sertão 99170-000, RS, Brazil; rosangela.poletto@sertao.ifrs.edu.br

**Keywords:** poultry, poultry farming, cage free, free range, anomalous behavior, animal welfare

## Abstract

**Simple Summary:**

Simple Summary: Recent studies have delved into the adverse phenomenon of smothering in cage-free laying hen-rearing systems, challenging the traditional notion of this behavior as “natural” or the result of hysteria among birds in the flock. This work identifies smothering as a detrimental, abnormal behavior with significant economic repercussions for poultry farming. Through a comprehensive literature review and bibliographic mapping, combined with consultations with poultry farmers via extension services and rural technical assistance, this study illuminates the environmental triggers of smothering behavior. The investigation reveals that factors inherent to the birds’ rearing environment precipitate this behavior, underscoring an urgent need for detailed, focused research into avian behavioral physiology. The objective is to unravel the complex interplay between production systems, animal welfare, and their economic implications on poultry operations. This study not only advances our understanding of bird behavior in intensive production contexts but also offers valuable insights for improving welfare standards and economic outcomes in the poultry industry.

**Abstract:**

The proliferation of rearing systems providing opportunities for birds to engage in natural behaviors can trigger behavioral repertoires that when not manageable compromise animal welfare and the economic viability of the flock. Smothering in laying hens has long been perceived as “natural” or the result of hysteria among birds in the flock. However, the current literature has recognized smothering as an abnormal outcome with the potential to result in significant losses in cage-free poultry systems. Recent studies have specifically aimed to categorize the organization of smothering behavior and highlight its potential causes and consequences. In this study, literature review and bibliographic mapping, drawing on published articles and engagement with poultry farmers through extension and rural technical assistance, were employed. The findings indicate that smothering is a behavior triggered by factors related to the environment in which the laying hens are kept. This study concludes that there is a critical need for more rigorous and detailed research to elucidate the nuances of avian behavioral physiology and assess the impact of production systems on animal welfare and the economic impacts on the flock. This research contributes to a deeper understanding of bird behavior in high-production environments and provides practical insights for the poultry industry.

## 1. Introduction

Since the Industrial Revolution, there has been progressive growth in poultry production systems, primarily aimed at maximizing economic returns. In parallel, global awareness of the importance of animal welfare practices has emerged, reflected in increasingly stringent policies aimed at prohibiting poultry breeding systems that neglect ethical principles and respect for animal integrity. Such policies are increasingly valued by consumers, directly influencing market selectivity. Initiatives aimed at promoting animal welfare and eradicating production methods that violate established ethical norms have gained prominence and momentum on the international stage. This movement reflects an upward trend in the implementation of management strategies that adhere to strict ethical criteria, resulting in direct pressure on the industry to adopt sustainable and humane practices.

Consequently, egg-laying hen husbandry systems have had to adapt to new requirements, with the intent of relocating the birds to environments that enhance animal welfare [1]. Examples of incentive policies include the European Citizens’ Initiative “End the Cage Age”, which gathered over 1.4 million signatures advocating for the gradual abolition of cage use in animal husbandry and the revision of current EU legislation [2].

There is an escalating demand for eggs produced by cage-free birds, which is anticipated to increase further in the upcoming years [3], resulting in the adoption of systems that optimize rearing concerning animal welfare [4]. The term “cage free” has come to be used to describe cage-free laying hen-rearing systems. While it is difficult to determine exactly when and where the term first came into use, concern for animal welfare and sustainable farming practices gained prominence in the last decades of the 20th century. 

However, it is known that when transitioning to systems that offer a non-sterile environment, natural and innate behaviors are expressed, such as those inherent to the animals that they are typically motivated to perform [5]. Notable examples of positive behavioral repertoires expressed by hens include nesting [6], perch seeking and roosting [7], contrafreeloading and forage-related behaviors [8], vocalization [9] preening [10], and others.

While cage-free systems on a large commercial scale provide beneficial behaviors for the birds, they also introduce challenges to farm managers and poultry producers [11]. In cage-free environments, laying hens demonstrate expanded behavioral repertoires and increased mobility. However, these systems may also correlate with a heightened incidence of detrimental behaviors compared to traditional caged systems [12]. Drastically reducing animal welfare, these practices lead to productive losses, demotivate the poultry chain, and result in significant economic losses [13]. One of the prominent abnormal behaviors associated with and triggered by extrinsic factors in rearing is smothering in laying hens [14,15]. Abnormal behaviors represent behavioral patterns that deviate significantly from what is considered typical or natural for the species. These behaviors often arise as a response to stressful or inadequate environments, are indicative of compromised welfare, and can be influenced by factors such as confinement conditions, improper management, environmental deprivation, and psychological or physical stress [16].

Smothering is seen as a bottleneck in the production of cage-free eggs in laying hen farming, as it not only reduces animal welfare but also leads to economic losses for poultry farmers. There is a scarcity in the scientific literature related to the problems caused by abnormal behaviors in cage-free birds. However, scientific reports from the United Kingdom [14,17,18] document investigative losses since the legislation banning conventional cage-rearing systems began in 2012 [19].

The abnormal behavior of smothering in laying hens necessitates comprehensive scholarly investigation to inform the scientific community effectively. It is essential to compile a data set directly related to production for those employing cage-free and free-range systems, particularly in response to current global demands. The primary objective of this literature review is to elucidate the complexities between the occurrence of smothering in cage-free production systems and the multifactorial elements that induce it, alongside their associations with behavioral repertoire, density, and physical environment.

## 2. Behavioral Dynamics in Herding and Group Composition Analysis

For a comprehensive understanding of the development of abnormal behaviors such as smothering, it is crucial to first revisit concepts such as the dynamics of herd behavior and the composition of groups formed by laying hens. Therefore, in this section, a detailed analysis of these aspects will be conducted, offering the reader a deeper insight into the subject.

The ontogenetic development related to natural classification and flocking behavior is depicted from the basic principles of behavioral ecology [20]. Grouping serves to benefit and enjoy conveniences that the action of more than one individual can bring. For instance, in the proliferation of a species when a male decides to mate with a female to produce offspring, they have a mutual interest in the survival of their progeny because both gain advantages if the offspring survive [21]. However, success will only be achieved if the aggregation is sufficiently organized, which consequently results in safety and alertness to danger or defense in case of attack, leading to a lower risk of predation and reduced energy costs.

Groups are formed through self-organization mechanisms and phenotypic variation [22]. The behavioral basis of hierarchical positions and chain organization stems from the heterogeneities that make up social organization, distributing submission, dominance, and hierarchy [23]. This is a social phenomenon observed in various species, from mammals to birds, and plays a fundamental role in structuring social interactions and resource allocation within groups [24].

Hierarchy refers to the social organization that individuals occupy in relation to different positions on a scale of power, while dominance is related to an individual’s ability to control or influence other group members [24]. These social patterns are often established through behavioral interactions, such as displays of aggression, specific body postures, cognition, cooperative relationships, and vocalizations [9]. Hierarchy and dominance play fundamental roles in the allocation and access to resources, such as food, water, environmental enrichments, resting spaces, and even the choice of reproductive partners, directly impacting the reproduction and survival of the species [24]. Furthermore, they are also responsible for the influence that some birds have over others, directly impacting the interaction and social behavior of the birds [25].

The species Gallus gallus domesticus, despite having undergone significant genetic modifications for domestication and subsequent selection of productivity-related traits [26], still carry in their genetic expressions ancestral behaviors originating from their earliest descent from the wild red junglefowl, Red Jungle Fowl [27]. These behavioral repertoires in current lineages are less expressive than those of their ancestors, yet they are of great importance for cognitive and productive development [28].

Domesticated birds have hierarchical social organizations [29]. Interaction among birds can be observed in small groups of 6 to 10 individuals [30]. The interrelation can be affected by a range of predisposed factors within the housing environment, as well as the arrangement of resources [31]. The more sterile the housing, the greater the limitation on birds in exercising behavioral repertoires [32]. The rearing of cage-free birds allows for the expression of natural and innate behaviors related to animal welfare, which the conventional cage housing system does not permit [33]. However, the possibility of birds exercising their natural behaviors, especially when raised free-range within a commercial system, triggers challenges related to the rearing system itself, many of which are associated with behaviors undesirable for production [34].

Birds are gregarious animals with varying levels of hierarchy. When given the opportunity to live in groups, they exhibit behaviors from their ancestry [35]. Restriction or limitation of physical space, pertaining to the infrastructure of housing (barrier effect) in cage-free rearing systems, can negatively impact social behavior, harming the group’s animal welfare [36,37].

## 3. Physiology of Fear and Aversion vs. Abnormal Behaviors

Fear is defined as the sensation of facing something harmful and frightening to the animal [38]. Genetic composition, negative experiences encountered from an early stage, and associative learning experiences are elements that have the potential to impact the level of fear in an animal [39]. Consequently, an animal’s apprehension represents the likelihood of reacting with fear to various potentially dangerous stimuli [40]. According to the same author, the physiology of fear in chickens involves an interaction between the central and peripheral nervous systems. When exposed to stimuli representing potential threats, domestic birds experience a physiological response that includes activation of the sympathetic nervous system. This triggers the release of stress hormones, such as corticosterone and adrenaline, which directly influence heart rate, blood pressure, and metabolic response [41]. Additionally, there is modulation in the endocrine system, with the release of neurotransmitters like norepinephrine, playing a crucial role in the manifestation and regulation of fear responses [40].

Fear in birds triggers a complex interaction between their physiology and behavior, playing a crucial role in adaptation and survival. When exposed to stimuli perceived as threatening, birds exhibit behavioral responses such as rapid evasion or seeking shelter [42]. Additionally, a potential threat can also induce immobility, serving as a survival strategy to avoid detection by predators [43]. This behavior, known as “tonic immobility” or “fear-induced immobility”, involves reduced movement and maintaining a static position, often accompanied by attempts to camouflage within the environment [44,45].

The social environment can influence the behavior of laying hens, as the presence of fearful individuals can induce an increase in fear levels in other hens. This action triggers a domino effect, resulting in an overall increase in fear levels among the birds [6]. This phenomenon implies the presence of social influences in the manifestation of fear, highlighting that the behavior of one individual can impact the emotional state of others, creating a unique social dynamic in the laying hen environment [46].

There are studies that report the interference of stress throughout the lives of birds. If birds experience challenges related to fear when young, they will be more sensitive to dealing with stress in adulthood, resulting in disorder in group formation [47]. The use of electric wiring at various specific points within the facilities is an example of a stress-inducing interference. This device is used to prevent perching behavior on equipment, as well as on feeders and drinkers, to prevent nesting on the sides of the poultry houses, and to inhibit innate behaviors [48]. An electric current of sufficient magnitude, when passing through an organism, can cause adverse effects on an animal’s well-being, including painful sensations, suffering, injuries, or death. Such harmful effects can be deliberately employed and enhanced for control and containment purposes.

Behavioral restriction along with negative experiences generate frustration in the motivation of laying hens. This frustration has a negative impact on the routine management practices carried out within the poultry houses, making the birds more stressed and more susceptible to aversion to human presence [49] and to minor changes in routine. These stresses can lead the birds to engage in small escape flights within the poultry house, resulting in hysteria and subsequently abnormal behaviors. A deeper understanding of these physiological processes not only contributes to the comprehension of chicken behavior but can also guide management and animal welfare practices aimed at minimizing the impact of stress and promoting healthier conditions.

The perception of fear is closely linked to certain types of flocking behavior and is often an aversive response to external stimuli. Laying hens, being gregarious animals, flock together for self-defense and survival. When they encounter a new environmental condition or something that disrupts the normality of their rearing environment, they seek self-defense. Self-defense involves seeking the safety of the group and staying close to better respond to a threat and protect themselves. However, fear-driven flocking behavior is only accepted when the threat is scientifically proven, such as the fear of predators, whether aerial or terrestrial [43]. 

Moreover, aversive practices unrelated to birds’ routines also cause fear-driven flocking behavior, leading to aggregation behavior in large proportions. This is especially evident when there is no behavioral learning, that is, learned behavior. As a result of this, new elements in the routine, such as fans, adverse weather conditions, and disturbances around the barn, among others, contribute to this phenomenon. Understanding the interconnection between fear and behavioral physiology in birds is crucial not only for promoting animal welfare but also for improving management practices in breeding environments.

## 4. What Is Smothering Behavior in Laying Hens?

Clustering is among the most significant aberrant behaviors exhibited by birds, primarily in poultry farming systems devoid of cages [50]. Clustering is recognized as one of the predominant causes of mortality in free-range poultry farming systems (26% of occurrences), followed by cannibalism (6.2% of cases) [51].

For a long time, this behavior was described as a normal congregation, associated with high stocking densities (birds/m^2^), and erroneously correlated with overcrowded housing of broiler chickens (average of 18 birds/m^2^ or 35 kg/m^2^) [52]. However, for laying hens, it only gained more visibility with the increasing demand for eggs in cage-free systems, when this behavior became a challenge to animal welfare [53].

Huddling for an extended duration has been defined as an unusual escape behavior pattern triggered by fear (hysteria), followed by vocalization, evasion, and the act of hiding or clustering in the corners of the poultry house or beneath the feeders [54].

The characterization of huddling behavior also pertains to the phenomenon observed when birds gather in high densities at specific locations within their housing environment, as highlighted [17]. This behavior can result in risks such as fractures and fatalities due to suffocation or asphyxiation, with a higher incidence among birds positioned at the base and within the pile, as discussed [14]. Table 1 presents the ethological definitions of huddling, in chronological order of citation by authors.

The nature of smothering behavior has long been considered unpredictable [14,15,59]. However, [59] categorized it into three segments: fear-induced flocking (hysteria, panic), nest-box flocking, and recurrent flocking. Fear-induced flocking or panic flocking is caused by factors that scare the birds, such as the presence of predators, and is generally the leading cause of mortality [59].

Nest-box flocking occurs when many birds seek the same nest (typically in the corners of the shed, at the beginning or end of the row) for egg-laying, mainly in the early laying phase when the animals have not yet gained confidence in their own nest choice and prefer to follow more experienced ones [62]. Recurrent flocking is a phenomenon that, once it occurs once, tends to repeat itself multiple times, often associated with clustering on top of the litter or in the corners of the aviary, without an apparent cause or reason [63].

The study by [53] investigated the frequency of causes related to flocking events and categorized them into six categories, with particular emphasis on two of them: solitary birds involved in a wide range of activities, including sitting, standing, and pecking at objects (77.9% of flocking events); and mass movements of laying hens (7.6% of flocking events). 

In Figure 1, the initial phase involves birds beginning their approach by congregating closely, competing for specific physical spaces within the designated areas. Label “a” represents birds at normal density, with no inclination to cluster. However, in episode “b”, triggered by an extrinsic factor that arouses their curiosity, the birds start to gather in a manner that does not adversely affect their well-being or environmental conditions. Subsequently, this gathering leads to crowding, a precursor to the accumulation illustrated at point “c”.

The second phase is marked by the stacking of laying hens on top of each other, predominantly characterized by panting birds due to high population density in a confined, physiologically contested space. In the stage denoted by the letter “d”, the birds begin to form stacks, leading to a pyramid formation, where birds are positioned atop one another. Following this, in stage “e”, the birds at the base of the pyramid suffer from oxygen deprivation due to the weight of those above them. The third and final phase, indicated by the letter “f”, results in death by asphyxiation and fatal trauma for the birds located at the bottom of the stack [14,63].

## 5. The Relationship of the Physical Environment to Crowding

Having established the multifactorial relationships of piling in laying hens [7], it logically follows that the relationship with the physical production environment also contributes to this behavior. Factors such as available space, environmental enrichment, effects of ambient conditions (temperature, humidity, and lighting), resources like feeders, drinkers, and nests, as well as management practices, appear to be correlated [17,64]. Figure 2 highlights a graphical summary of the extrinsic factors directly related to the causes of piling.

### 5.1. Effects of Physical Space

The movement of birds through space and their interaction with available resources is particularly influenced by circadian rhythms and social behavior that leads to flock synchrony [63]. Birds naturally prioritize egg-laying in the morning, engage in active behaviors in the afternoon, and perch at night. This pattern, coupled with the simultaneous use of a single resource facilitated by social behaviors, can lead to the piling of birds if resources or available space are limited [66].

Regarding available space [67], a minimum provision of 600 cm^2^ per bird (0.06 m^2^ per bird) for flocks of 100 or more individuals is proposed to allow for static postures (such as standing) and dynamic behaviors (like wing flapping). In contrast, the Humane Farm Animal Care standards vary between 0.09 m^2^ to 0.14 m^2^ per bird depending on the housing and rearing systems used.

It is acknowledged that the homogeneous distribution of animals in different areas of the barn is critical [68]. Additionally, it is estimated that birds use at most 80% of the total available space [69]. Studies show that larger enclosures enable greater dispersion of animals and a higher quantity of exploratory movements by laying hens and broilers. This indicates improved welfare and, potentially, a reduced likelihood of piling.

Leone and Estevez (2008) [37] demonstrated that there is a higher flow of birds (ranging from 9 to 41 birds/m^2^) in certain areas of the barn, especially in large flocks with more extensive housing spaces. This suggests that piling in relatively small areas of the barn is more likely to occur in larger flocks. This supports the hypothesis in [17], which determined that piling is more frequent as group size and population density increase. This is because a larger number of birds can pile up, and birds have social stimuli and preferences more likely to result in abnormal behaviors.

Piling with a greater number of animals is also suggested in larger flocks (>12,000 birds compared to flocks of 6000 birds or 6000 to 12,000 birds), regardless of the available space [61]. The farming system was a factor considered [70] in relation to piling, demonstrating the propensity of the behavior in slats/litter and aviary systems to be relatively high when compared to systems offering outdoor spaces (free-range areas), which was considered moderate. Therefore, the available area, the size of the lots, and consequently, the occupation density are crucial factors in the establishment and control of piling.

### 5.2. Effects of Available Resources and Environmental Enrichment

When birds are housed in groups, the structure of their living environment can impact their behavior [71] through movement freedom (barrier effect) [72] and the level of welfare it provides. Weeks and Nicol (2006) [73] determined that birds tend to avoid sterile environments due to their exploratory nature, actively working to access nests, perches, and foraging materials, indicating that the resources provided also affect their behavior. Collins et al. (2011) [64] pointed out that the motivation for piling can be driven by limited access to resources such as feeders, drinkers, nests, and perches. The importance of ensuring access lies in the fact that birds often synchronize their behaviors (wing flapping, dust bathing, and laying), which can lead to overcrowding at specific locations/resources in the barn, causing piling [18,63].

Gregarious nesting, the preference of birds for an occupied nest over an unoccupied one [74], is a classic example of chicken social behavior that can lead to piling [42] at specific locations/resources. Barrett et al. (2014) [14] demonstrated that, out of 12 egg farm managers interviewed in the UK, at least 30% recorded piling events in nest boxes. Furthermore, a preference for nests located in the corners of the facilities, at the beginning or end of the row of nest boxes, was observed [15,62].

Regarding the development of a social hierarchy, dominant birds primarily access available resources, potentially leading to piling as subordinate birds attempt to access them [18]. Therefore, providing an appropriate quantity and maintaining efficient resources such as feeders, drinkers, and nest boxes, and even the availability of additional quantities of these resources can help reduce piling behavior [7].

Conversely, a lack of environmental enrichment can also be an influencing factor for piling, causing animals to develop a socially oriented focus due to a lack of competitive stimuli [65]. Additionally, a lack of a complex environment can exacerbate animals’ fear response when exposed to sudden and unexpected stimuli [75], leading to behavioral disturbances such as piling.

Environmental enrichment can enhance birds’ welfare through the expression of feelings of pleasure, interest, and a sense of control [76], as well as improve the use of resources [77] and available space [78]. Dumontier et al. (2022) [79] also observed that exposure of laying hens to environmental enrichment before and during the laying period reduced fear in response to a new object introduced into the system. Moreover, [80] found that complex environments have a positive effect on egg weight and total mass, as well as improvements in feed conversion rate and reduction in bird mortality during the laying period.

The use of materials like sand, wood shavings, rice hulls, and long-cut straw can optimize dust bathing and foraging behaviors [81]. Additionally, the availability of stones or polystyrene blocks, and devices with ropes or hanging beads can aid in the expression of pecking behavior [82]. Perches and elevated platforms are also used to offer the possibility for animals to explore and exercise through flight [83]. Finally, according to [84], providing a visually, audibly, olfactorily, and tactilely enriched environment seems to improve the welfare of housed animals and, consequently, productivity and profitability in the system. Given the responses related to the production environment and piling, it is evident that various materials can be introduced into the production environment to reduce the possibility of piling occurrence.

### 5.3. Effects of the Environment—Temperature and Lighting

The thermo-physical conditions of the production environment (ambiance), encompassing variables such as temperature, relative humidity, lighting [63], and even solar radiation [65], can promote piling by making certain areas of the barn more attractive [18]. 

According to [85], temperature can affect animal behavior. Birds under thermal stress tend to consume less food, drink more water, and increase movement, potentially exacerbating innate behaviors like dust bathing [86] and possibly abnormal behaviors like piling, which can be caused by attraction to warmth or aversion to cold [17]. 

Bright and Johnson (2011) [59] reported that producers identified daily temperature fluctuations as one of the probable causes of piling, with a potential for suffocation in laying hens. Moreover, changes in grouping behavior, dependent on barn temperature, were observed in Leghorn chickens [86]. Herbert et al. (2021) [87] also noted an association between the temperature range in the barn and the higher number of birds counted in a piled stack, describing the event as a cause or effect of piling and suffocation. This coincides with the findings in [88], which highlighted temperature variation due to reduced shading in poorly distributed areas, especially on sunny days, as well as opportunities for dust baths and other elements that draw birds to specific locations within the barn, as conducive to piling behavior.

When measuring temperature in the corners of the barn using thermographic cameras, no interference was found in the effect on the characteristics leading to clustering behavior [50]. Similarly, simulations with thermal plates raising the temperature in the environment did not result in behavioral changes [53]. Nonetheless, given the discrepancy in data regarding the relationship between temperature and the piling of laying hens, the authors suggest more in-depth studies on this variable [17,18,63].

Beyond temperature, it is known that illuminance and lighting regimes (hours of light, wavelength, and light distribution) affect bird behavior, especially regarding reproductive functions and biological cycle, hostility, and space utilization [68]. For example, ovarian weight and the number of mature follicles respond positively to increased light intensity, which directly affects laying [89]. Another example is the animals’ preference for lying in areas with low light intensity, leading most nest boxes to have no light inside [90], making them more attractive for laying.

Figure 3 illustrates a behavioral phenomenon observed in laying hens, where significant piling of individuals is induced by the presence of an artificial light source. This behavior can be attributed both to the visual stimulus provided by the light and the search for warmth radiated by the lamp.

Although [17] noted that phototaxis (attraction to light—positive phototaxis or repulsion to light—negative phototaxis) has not been demonstrated in laying hens, observations of piling initiated by birds’ attraction to fragments of external light have been reported [91]. Uneven beams of natural light may enter through openings in curtains or gaps in barn walls [65], leading birds to cluster either in search of a heat source or out of curiosity.

Solar light beams account for about 2.6% of piling causes [53], while patches caused by artificial lamps represent 0.9% of piling causes [18]. Figure 4 illustrates a real situation where subgroups of laying hens compete for a space enriched from the birds’ perspective. In the depicted figure, it is the beam of solar light coming from outside that falls inside the barn, leading the birds to cluster to the point of piling up.

Birds also show preferences for different types of artificial light sources. Chickens exposed to artificial lighting prefer fluorescent lighting over incandescent [92], and fluorescent over LED (light-emitting diode) lighting [93]. Meanwhile, chickens exposed to incandescent artificial light and natural light develop a preference for natural light [94]. Furthermore, certain colors have the potential to evoke specific responses/behaviors in birds. Blue can induce adverse reactions, while red is perceived as a danger signal, and yellow (and light colors in general) is attractive to animals [68]. These two examples of preferences (light sources and colors) demonstrate that birds will spend more time in their preferred areas in the barn, such as light focal points, nests, and feeders, which can lead to overcrowding and, consequently, piling of animals in certain spaces [17]. 

Additionally, it is important to consider that birds possess tetrachromatic vision, meaning they have four types of photoreceptors enabling them to detect blue, green, and red light, and shorter wavelengths (100–400 nanometers), such as ultraviolet light [95]. This means that these animals not only have better color acuity but also can identify wavelengths invisible to humans [96]. Therefore, in addition to considering the human-visible light spectrum (400–700 nm), it is necessary to contemplate that birds may be attracted or repelled to focal points by lights or colors invisible to us, potentially causing piling with no apparent motivation [17,66]. Again, this highlights a set of factors that, combined with those previously discussed, could increase the necessary management practices or indicators to reduce piling in cage-free poultry systems.

### 5.4. Effects of Human–Animal Management and Interaction

Good management practices in cage-free farming systems present more challenges compared to battery cage systems. Handlers spend more time dealing with a greater number of variables, such as opening/closing gates for paddocks, assisting animals in using unoccupied nests (especially in the first weeks of laying), gathering floor eggs, and dispersing animals when they exhibit variations in normal behavior [17].

Additionally, increasing scientific evidence suggests that interactions between handlers and animals can significantly affect well-being and, consequently, the behavior exhibited by the birds, especially during episodes of fear and stress [97,98]. In cage-free laying hen farming, the interaction between humans and animals is vital for the birds’ welfare and productivity. Hemsworth and Coleman (2010) [98] highlight that regular, gentle handling greatly reduces stress and fear. This leads to better health and productivity in the birds. The authors of [17] note that birds can find human presence in the barn either comforting or stressful, depending on how they interact.

Laying hens are creatures of routine and can be conditioned to behavioral repertoires that benefit the individual or the group as a whole [77]. For instance, it is known that birds consume more food 2 to 3 h before the dark period and before oviposition starts [99]. Improper feeding management, such as inadequate feeding times (especially during laying) and insufficient food availability throughout the day, can lead to competition among birds, which becomes a stressor that can propel abnormal behaviors, like piling [100]. Additionally, dietary restrictions, changes in feed composition, and food attractiveness can be stress-inducing triggers [101].

Handlers need to be attentive to the animals throughout the housing period; however, the beginning and peak of laying are critical phases for welfare and the development of piling behaviors. This occurs for three reasons: during the initial laying period, animals are not yet confident in their ability to choose the appropriate nest for egg-laying, so they follow dominant or more experienced birds to the same nests, which can lead to piling in the nesting boxes [62]. During peak laying periods, there is high demand for nests, especially those located in corners, at the beginning or end of the row, which can also lead to piling in nesting boxes [68]. Additionally, gregarious nesting, previously discussed, can also lead to piling in nesting boxes [15]. 

It is also important to consider that 85% of the total laying for a flock occurs in the morning, with a significant demand for adequate available space. Therefore, the handler’s role includes guiding animals to an available and suitable nesting box, preventing competition for resource use, and helping to prevent piling [65]. Nonetheless, it is crucial to ensure that facilities are equipped with an ideal number of nests for the housed flock; otherwise, it will be another factor driving piling.

Production systems that offer birds the opportunity to access outdoor areas on sunny days, such as free-range, backyard, and organic systems, can pose significant challenges to animals on cloudy, rainy days, or when environmental conditions and soil quality do not permit outdoor access. Due to the natural behavioral conditioning from routine outdoor access perceived as a positive reinforcement, birds tend to pile up at the gates when prevented from accessing their exit at the usual management time [102]. To prevent any harm, it is necessary for handlers to disperse the animals.

## 6. Bibliographic Mapping

A comprehensive review of the global literature on piling in laying hens was conducted using Scopus (Elsevier data). The programming code utilized in this research is (TITLE-ABS-KEY (smothering) OR TITLE-ABS-KEY (pilling) OR TITLE-ABS-KEY (crowding) AND TITLE-ABS-KEY (heavy AND broiler AND matrix) OR TITLE-ABS-KEY (laying AND hen)) AND (LIMIT-TO (SUBJAREA, “AGRI”) OR LIMIT-TO (SUBJAREA, “VETE”)) AND (LIMIT-TO (DOCTYPE, “ar”) OR LIMIT-TO (DOCTYPE, “re”)) AND (LIMIT-TO (LANGUAGE, “English”)).

Keywords were selected and included according to researchers in the field of animal behavior and the objectives of this study. The applied filters were: (1) language (articles only in English); (2) field (Agricultural and Biological Sciences and Veterinary); and (3) article type (search limited to articles and reviews), excluding conference abstracts and pre-prints.

The information included keywords and keyword indices, which were exported in CSV format. The retrieval date was January 18, 2024. VOSviewer (version 1.6.18) was used to map the keywords (Van Eck and Waltman, 2010, 2013, 2017). The circle size in VOSviewer positively correlates with the occurrence of the keyword. The color intensity of the line connecting the words is more vibrant if the word is commonly found across different studies. If the connection is weak, the color will be more transparent [103].

In Figure 5, the main scientific insights related to the words linked with the piling of laying hens are represented. Each color shade represents a cluster, i.e., a group of words that have affinities [104] and are closely linked to the birds’ clustering behavior. This figure was utilized to centralize terms, with the most evident and central description being “animal behavior”, followed by “animal welfare”, and “female” (because piling is a characteristic predominantly observed in females). The further from the center, the less scientific evidence was presented and correlated with the term piling. However, these are insights from research that could be further explored on the issue.

Upon conducting a bibliometric analysis of published works utilizing the term “piling” in avian studies, the search yielded a constrained scope of the literature. This examination revealed a publication timeline spanning from 1977 to 2023, with a notable peak in scholarly activity observed in 2016, as depicted in Figure 6. This temporal distribution of publications provides a critical insight into the research trends and evolving focus within the field of avian behavior studies, particularly concerning the phenomenon of “piling” in birds.

Figure 7 shows a bibliometric analysis focusing on academic production related to the term “piling” in laying hens in which a significant predominance of publications from the United Kingdom is observed. This concentration can be partly attributed to the implementation of the European Union Directive 1999/74/EC [19], which establishes minimum standards for the protection of laying hens, particularly regarding the prohibition of conventional cage systems. This legislation has significantly influenced poultry management practices and, consequently, stimulated research in this area in the UK.

On the other hand, there is a noticeable gap in publications from countries with high egg production, such as China, which leads global production with 38% of the total, which is equivalent to 586 billion eggs. China is followed by India, with a production of 122 billion eggs, and Brazil, which holds the fifth position globally with 58 billion eggs [3]. This discrepancy suggests a significant opportunity to increase research and understanding of the practices and challenges associated with egg production in these high-production countries.

Furthermore, there is a growing trend in scientific investigation related to alternative poultry farming systems, as well as the behavioral and welfare consequences associated with different management methods. This trend is evidenced by the expansion of studies exploring the impact of cage-free and free-range systems on birds’ behavior and health. This increase in research volume parallels the growing global awareness of animal welfare and consumer demands for more ethical and sustainable production practices in poultry farming.

## 7. Conclusions 

In accordance with the survey conducted in this article, the implications and challenges of piling in laying hens were assessed, highlighting the critical importance of appropriate management practices and optimized environments. However, beyond environmental and management interventions, an auspicious and necessary direction emerges for future research and practical applications. Further understanding of the neurophysiological basis of natural behaviors triggered by adverse conditions that lead to piling is urgent for sustaining the welfare and success of cage-free hen-laying systems.

To tackle the challenges of piling, it is crucial to evolve poultry genetics alongside field practices and handler care. The goal is to create strains that are less reactive to different environmental stimuli and stress. This approach aims to reduce negative behaviors and improve quick adaptation to changing conditions. The crowding of laying hens directly influences management systems and practices, as well as aspects related to the physical environment of the facility.

Finally, as we advance in this direction, it is crucial that the scientific community, poultry farmers, and industry professionals continue to collaborate. Together, they can develop strategies that align animal welfare with the demands of poultry production, ensuring a more sustainable and ethical future toward cage-free laying hen farming.

## Figures and Tables

**Figure 1 animals-14-01518-f001:**
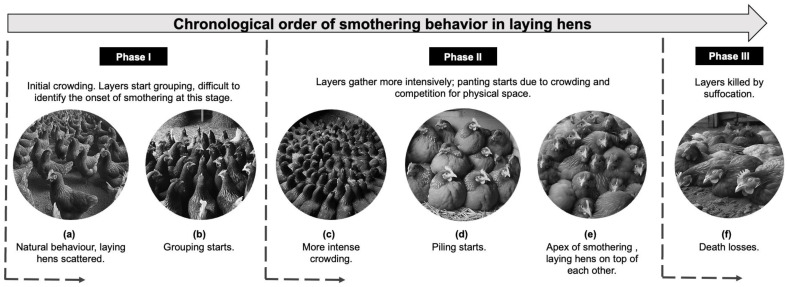
Chronological order of the piling behavior in laying hens (**a**–**f**). In Phase I, the birds initiate clustering behavior, but without competing for physical space. Phase II sees an escalation in clustering intensity, leading to the onset of suffocation. In Phase III, the weight of the stacked birds results in fatalities. Source: Image generated by artificial intelligence.

**Figure 2 animals-14-01518-f002:**
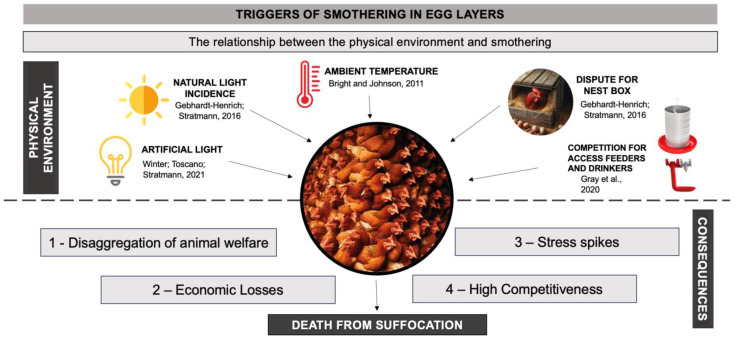
Relationship between physical environment and piling in birds: causes and consequences [17,18,59,65].

**Figure 3 animals-14-01518-f003:**
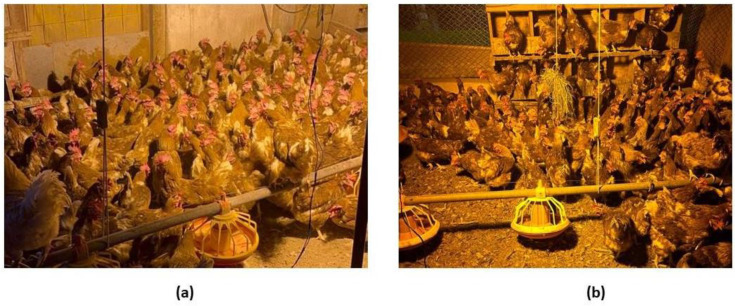
(**a**) Piling behavior near the walls of the barn and (**b**) in the corners of the barn influenced by a beam of artificial light. Source: Author.

**Figure 4 animals-14-01518-f004:**
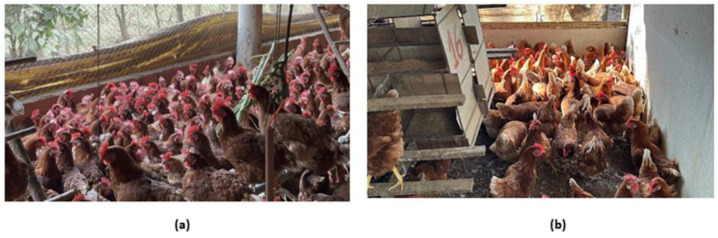
(**a**) Piling behavior near the walls of the barn and (**b**) in the corners of the barn influenced by a beam of solar light. Source: Author.

**Figure 5 animals-14-01518-f005:**
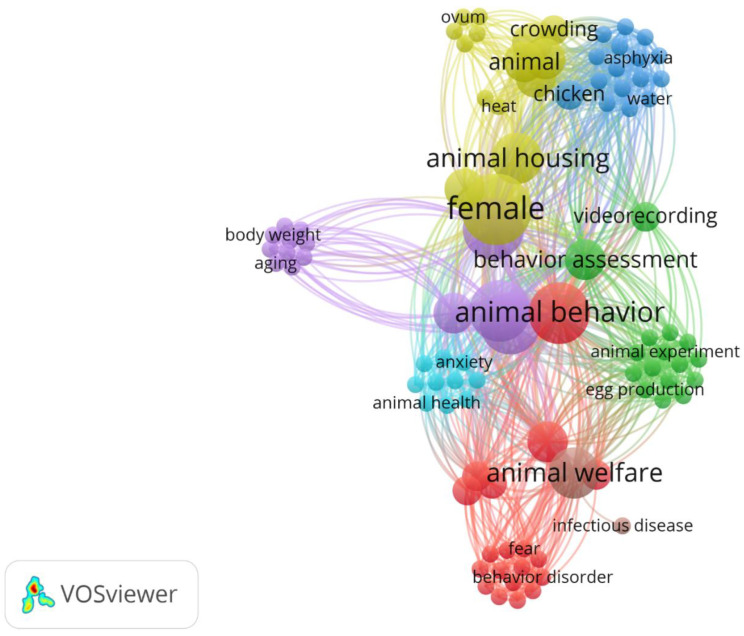
Bibliographic coupling analysis for keywords related with piling in laying hens. Different colors identify the clusters of keywords most prominent across the publications analyzed.

**Figure 6 animals-14-01518-f006:**
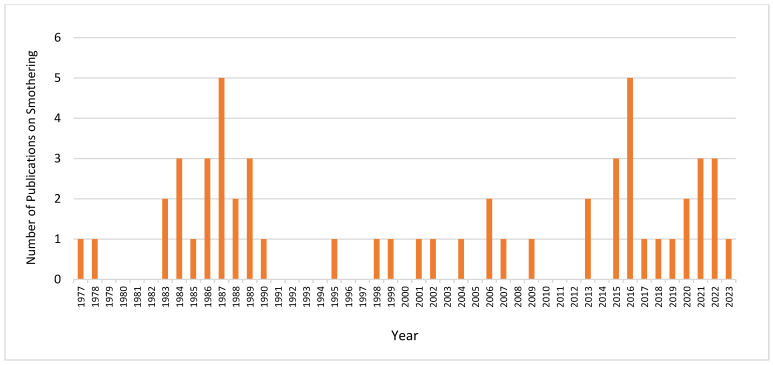
Timeline of publications on smothering found in the bibliographic database from 1977 to 2023 provided by Scopus, Elsevier (2024).

**Figure 7 animals-14-01518-f007:**
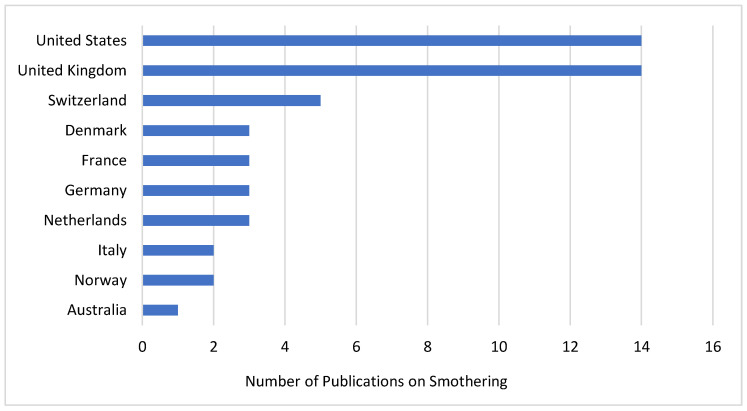
Countries with the highest rates of academic publications on smothering are found in the database provided by Scopus, Elsevier (2024).

**Table 1 animals-14-01518-t001:** A bibliographic survey of historical citations of huddling behaviors reported by various authors.

Authors	Reference	Background	History	Category
Hart (1939)	[55]	The physiological effects of disordered clustering are uniformly detrimental.	Clustering is treated as disordered aggregation.	-
Sanger VL (1962)	[54]	Unusual escape behavior triggered by fear (hysteria) and the tendency to hide in corners or cluster beneath feeders.	Anomalous behavior	Laying hens
Mills (1990)	[56]	Asphyxiation is a significant economic consequence of panic and hysteria in domestic poultry.	Unsatisfactory outcome of hysteria.	Laying hens
Michael C. Appleby et al. (2004)	[57]	Clustering is primarily caused by limited space, but it becomes more pronounced in confined environments and small groups.	Clustering is depicted as unsatisfactory, associated with high density.	Broiler chickens
McMullin (2006)	[58]	Clustering in a specific area, with those in the center at risk of death due to lack of oxygen and/or excess carbon dioxide.	Triggered by unusual activity.	Laying hens
Bright and Johnson (2011)	[59]	Unpredictable behavior is unknown in the free-range egg industry.	Clustering due to an extrinsic factor, likely a fright.	Laying hens
Rabiu Mohammed et al. (2015)	[60]	A case of non-infectious mortality.	-	Quails
Rayner et al. (2016)	[15]	Three categories. These were (1) panic suffocators (PS), (2) nest box suffocators (NBS), and (3) recurrent suffocators (RS).	-	Laying hens
Gray et al.	[17]	Described by severe clustering at points in the poultry house, leading to piles.	Smothering as pyramid formation.	Laying hens
Winter et al. (2021)	[18]	Smothering is typically initiated with chickens being influenced by the behavior of their companions and by high concentrations of animals in certain locations.	-	Laying hens
Winter et al. (2022)	[50]	Huddling was often observed outside of common interaction hours and appeared to be more likely around noon and when the chickens engaged in various behaviors in the corners of the poultry houses.	-	Laying hens
Armstrong et al. (2023)	[61]	In all observations, huddling behavior occurred, regardless of a history of asphyxiation. This suggests that such behavior may be more common than commonly believed.	-	Laying hens

## Data Availability

Data supporting reported results can be found with the corresponding author, including datasets analyzed and generated for the study.

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
