# Peer review of "Laying Hens: Why Smothering and Not Surviving?—A Literature Review"

_animals, 2024, doi:10.3390/ani14111518_

Round 1
Reviewer 1 Report
Comments and Suggestions for Authors
I am expressing my gratitude for the opportunity to review the article titled "Laying hens: Why smothering and not surviving? A literature review."
This is an important issue in the keeping of laying hens in cage-free conditions.
However, I also have some critical points and concerns about the paper:
1) lines 57, 65 and following: A definition or framework conditions of "cage-free" in different countries is missing regarding effects on the behaviour of laying hens. There are major differences in the realisation of the term "cage-free" regaring conditions for layers.
2) Please provide an overview of the, as expected, very different framework conditions of cited studies (for example in the form of a table), such as housing system (floor housing/ aviary), herd size, age of animals, genetics, stocking density, access to free range, rearing conditions of the pullets, nests (ratio of number of animals, design, negative influencing factors...), the presence e.g. of live cables in the corners of the stables,...
The differences may be of great importance for the evaluation and interpretation of the results.
3) Line 75 and following: How is “stress” defined or what exactly is meant by the term "stress"? For example, stressful (environmental) factors that the animal can no longer compensate for?
4) In addition to the influencing factors for piling of hens listed in the manuscript: Is there a relevance of further parameters as for example the toleration reflex of sexually mature hens?
5) Line 270, figure 2: ambient temperature – I would have expected an influence of enthalpy instead of temperature alone?! Please refer to this aspect.
6) In addition to no. 5: I would have expected an influence of further parameters of the barn climate such as the concentration of ammonia. Please refer to this aspect.
7) line 488 (concluding remarks): Please present the relevant results of the review concisely.
As a result, I recommend publishing after minor revisions.
Author Response
Dear Reviewer,
We are immensely grateful for the impeccable effort you have put into reviewing our article. We would like to commend your professionalism and depth of knowledge, as well as the objectivity related to the review topics. We also appreciate the insights you provided, which have opened up avenues for us to consider and shape future research. Please accept our sincere thanks.

Reviewer 2 Report
Comments and Suggestions for Authors
This article mainly discusses the problem of smothering in cage-free laying hens induced by the accumulation. Based on a thorough review of historical documents, the causes and characteristics of huddling are introduced, especially the effects of the physical environment and management. However, for breeders, the problem of smothering caused by stacking in cage-free laying hens is not an extremely important problem that significantly affects breeding efficiency, so the relevant literature reports are limited. The physiological mechanism of stacking has not been studied. Therefore, the present review is of little significance. Future research is needed to understand further the neurophysiological basis of triggering adversity conditions to breed strains less responsive to different environmental stimuli and stresses to reduce stacking-induced smothering.
Comments on the Quality of English LanguageOK.
Author Response
Dear Reviewer,
Thank you for your comments. We would like to highlight the importance of your review on our article. Your insights have not only helped us improve the manuscript but also opened new perspectives for us. Please see the attached document for the revised version.

Reviewer 3 Report
Comments and Suggestions for Authors
The manuscript takes a new look at the current issues of poultry rearing technologies, among which is the problem of bird smothering, which arises even after the application of friendly poultry farming technologies, such as the cage-free system, which allow to create conditions for birds to engage the natural behaviors and have been a major aspiration until now. A number of scientific studies in this field were reviewed and an attempt was made to summarize them.
While reading the manuscript, I sometimes miss a clearer presentation and it is difficult to understand which summarizing statements are made by the authors themselves, and where they repeat only the statements of cited authors.
Anyway, I think the manuscript can be published in a journal
Author Response

(The authors gave the same response as above.)

Round 2
Reviewer 2 Report
Comments and Suggestions for Authors
Ok.
Comments on the Quality of English LanguageOk.